# Behavior of Engineered Cementitious Composites (ECCs) Subjected to Coupled Sustained Flexural Load and Salt Frost

**DOI:** 10.3390/ma16010165

**Published:** 2022-12-24

**Authors:** Yonghao Li, Ning Zhang, Renjuan Sun, Yanhua Guan, Lemin Liu, Changjin Tian, Yifeng Ling, Hongzhi Zhang, Branko Šavija

**Affiliations:** 1School of Qilu Transportation, Shandong University, Jinan 250002, China; 2Shandong Hi-Speed Engineering Test Co., Ltd., Jinan 250002, China; 3Shandong Expressway Infrastructure Construction Co., Ltd., Jinan 250002, China; 4Suzhou Research Institute, Shandong University, Suzhou 215021, China; 5Microlab, Faculty of Civil Engineering and Geosciences, Delft University of Technology, 2628 CN Delft, The Netherlands

**Keywords:** engineered cementitious composites (ECCs), salt frost, sustained flexural load, chloride penetration depth, free chloride profile

## Abstract

The performance of engineered cementitious composites (ECCs) under coupled salt freezing and loaded conditions is important for its application on the transportation infrastructure. However, in most of the studies, the specimens were generally loaded prior to the freezing. The influence of sustained load was merely considered. To this end, four sustained deflection levels, i.e., 0%, 10%, 30% and 50% of the deflection at the ultimate flexural strength, and three salt concentrations (1%, 3% and 5%) were applied. Prior to the salt frost resistance test, the fluid absorption of ECC specimens under various conditions were measured. The changes in relative dynamic elastic modulus (RDEM) during the freeze–thaw cycles were captured. The depth and the content profile of free chloride were measured after the coupled sustained load and freezing and thawing cycles. It is shown that 3% NaCl solution leads to the largest deterioration in all cases. There is no visible flaking or damage occurring on the surface. The relationships between locally sustained flexural stress and RDEM loss and also locally sustained flexural stress and free chloride penetration depth were proposed and showed satisfactory results. It is concluded that when ECC is subjected to the FTCs under 1% de-ice salt solution, no depassivation of the steel is expected even under a large deflection level. In terms of 3% and 5% salt solution, the thickness of cover should be no less than 20 mm when a deflection level of 0.5 is applied.

## 1. Introduction

Engineered cementitious composites (ECCs) are short-fiber-based cementitious materials [1]. Sometimes they are called strain-hardening cementitious composites (SHCCs) [2] or ultra-high toughness cementitious composites (UHTCCs) [3]. ECCs usually consist of cementitious materials, fine aggregates and fibers. The cementitious materials generally include cement and fly ash [4,5]. It was shown that the incorporation of fibers resulted in excellent mechanical properties [6] and fatigue properties at high fatigue stress and strain levels compared to concrete [7]. Under the action of tension, ECC will exhibit pseudo-strain-hardening and multi-cracking behaviors. The deformation capacity of ECC under tension is much greater than that of plain concrete, which is nearly two orders of magnitude [7]. Furthermore, since ECC has a small crack width, it will have the ability to self-repair in the presence of water [8,9]. Due to its promising properties, ECC has a good potential for use in road engineering. Refs. [10,11] have applied ECC as an overlay on a steel bridge deck and focused on its performance in three-point bending resistance, under wheel axle load and transverse bending, respectively, and the results proved that ECC is eligible as a cover layer. Furthermore, ECC has been successfully used in a pavement overlay system to eliminate the reflective cracking from the subgrade [12]. It has been reported that, whether in static [13,14] or fatigue flexural conditions [15], the crack width of the ECC layer can be controlled effectively, generally less than 100 μm. So it is difficult for aggressive solutions to enter the interior of the ECC [16,17], resulting in less corrosion and increases in the structure life span.

Prior to the application on road engineering, the durability, e.g., salt frost resistance of ECC, has to be revealed. De-icing salt is usually adopted to reduce road icing and effectively prevent traffic accidents. However, this is a major cause of rapid concrete degradation, causing corrosion of reinforcement, internal cracking and surface scaling [18]. Physico-chemical effects will occur if concrete is in a freeze–thaw environment [19]. It is reported that the internal cracking reduces the dynamic modulus of material and leads to the increase in permeability and porosity [20]. In terms of surface scaling, the damage includes the clearing of fine chips or sheets of objects. It subjects the vulnerable body to moisture and aggressive minerals, threating the durability [21,22]. Glue-spall theory can be used to explain the scaling mechanism [23,24]. In this theory, the cause of frost salt scaling is the dehiscence of the ice/brine section. The liquid can be classified in three categories, depending on its solute concentration. The concentration of 1–3% is defined as the pessimum salt concentration in which the most severe surface scaling occurs. If the concentration is too low (<0.1%) or high (>10%), no ice cracking or scaling is expected. Furthermore, sodium chloride (NaCl) and other chlorides cause corrosion damage to transportation infrastructure [25,26].

Ref. [27] reported that sodium chloride (NaCl) is used on a large scale for de-icing. Understanding the behavior of ECC under sodium chloride de-icer is, therefore, a key issue for the application of ECC in transportation infrastructure. Liu et al. [28] studied the mechanism of solid waste on the salt freezing resistance of ECC, containing large amounts of fly ash. They showed that the salt-frost-caused degradation is more serious than tap water. Similar observations have been reported in [29]. The presence of fibers can significantly improve the resistance against the progress of scaling by bridging cracks. Van Zijl et al. [30] showed that ECC maintained durability even after freeze–thaw cycles with de-icing salts.

When it is in service, sustained load is applied on ECC. Cracks under sustained load are, therefore, present during the salt frost attack. However, there have been only a few studies taking sustained loading into account. Sahmaran and Li [31] evaluated the durability of ECC with cracks according to ASTM C672. Due to the de-icing salt, the various indicators of specimens are all maintained at an appropriate level even after 50 FTCs with 4% sodium chloride solution. The freeze–thaw durability of pre-cracked ECC in a chloride environment was reported by [32]. However, the research is limited to one stress level and a certain salt solution concentration.

Therefore, this study investigates the effect of salt solution concentration and stress level on the behavior of ECC. NaCl was used as the de-icer and liquids with three concentrations, i.e., 1%, 3%, 5%, were employed for the freezing and thawing cycles. Multiple sustained load levels were used, i.e., deflection at 0, 10, 30 and 50% of ultimate flexural strength. The relative dynamic elastic modulus (RDEM), the depth of chloride penetration and free chloride content were researched. Relationships between locally sustained flexural stress and RDEM loss, as well as locally sustained flexural stress and free chloride penetration depth, were proposed and showed satisfactory results at all salt concentrations.

## 2. Materials and Methods

### 2.1. Materials and Specimen Preparation

The materials used include PO42.5 grade cement, class F fly ash, quartz sand, superplasticizer, hydroxypropyl methyl cellulose-based viscosity modifying admixture (VMA) and polyvinyl alcohol (PVA) fiber. Superplasticizer used can reduce water consumption by nearly 40%, and the VMA was used to increase the consistency. PVA fiber was used to improve the ductility of the matrix. The cement and fly ash were locally produced in Shandong Province, China. The chemical compositions are shown in Table 1. The particle size of quartz sand was between 0.125 and 0.18 mm (80–120 mesh), with a fineness modulus of 0.8. PVA fibers (Kuraray Corporation, Kurashiki, Japan), a length of 12 mm and a diameter of 40 μm being used in this study. The mixture used was developed in a previous study [33] and is shown in Table 2. The mass of water was 0.26 (w/b) of the binder material and the volume ratio of fiber was 2%. The dimension of specimens was 400 mm × 50 mm × 25 mm. The stirring process and the curing conditions reference [34]. Prior to the coupled sustained flexural loading and de-icing salt scaling test, except the loaded surface and the one parallel to it, other surfaces of the specimens were sealed using epoxy.

### 2.2. Four-Point Bending Test

The test was first applied on the 28 day cured specimen to determine the flexural deformation capacity. The length of the support span and loading span were, respectively, 300 mm and 100 mm. Loading rate of specimen was 0.5 mm/min, this was achieved by the action of a universal testing machine. The mechanical properties were determined by averaging the results from at least 3 specimens.

### 2.3. Coupled Sustained Loading and De-Icing Salt Scaling Test

Chinese standard GB/T50082-20019 was utilized to evaluate the properties of the loaded ECC under salt freezing conditions. Multiple sustained load levels, namely 0, 10, 30 and 50% of the deflection at the ultimate flexural strength, were used. The construction diagram of the loading device is shown in Figure 1. The specimen was placed between the two loading frames. The supports point distance and loading span were 300 and 100, respectively. The load was applied by twisting the screw. The apparatus and the loaded specimen were placed into a tank with NaCl solution. Three solution concentrations (1%, 3% and 5%) were used.

Before the freezing and thawing cycles (FTCs), specimens were kept in the salt solution for 7 days. The weight of the specimen together with apparatus was measured before and after the saturation to calculate the absorption ability of the specimen. The liquid level was kept 5 ± 1 mm above the bottom of the specimen. The tank was put in a freeze–thaw testing machine with refrigerated liquids. The liquid level was kept 10 ± 1 mm above the bottom of the specimen height. The test was conducted every 12 h as a cycle. In each cycle, temperature was linearly reduced from 20 °C to −20 °C within 4 h, then maintained at −20 °C for 3 h. Afterwards, the temperature was changed linearly to 20 °C within 4 h and maintained at 20 °C for 1 h.

After every four FTCs, relative dynamic elasticity modulus (RDEM) was measured using ultrasonic waves technique [35]. The longitudinal RDEM was first measured at the two ends of the specimen. The middle part of the two loading points was the pure bending section in which the maximum bending moment was generated. The transversal RDEM was measured at positions A, B, C and D, as shown in Figure 2, which is corresponding to the planes under 0, 0.25, 0.5 and 1 of the maximum bending moment in the pure bending section, respectively. A is the support position, at which the bending moment is 0.

### 2.4. Chloride Penetration Depth Measurement

After 28 FCTs, the specimens were removed and cut along planes A, B, C and D (see Figure 2) without water. As a chloride color indicator, 0.1 mol/L AgNO_3_ solution was sprayed on the cut surface. In areas where the free chloride concentration is greater than 0.15 wt.% [36], white precipitation occurs. Otherwise, silver oxides with a brown color precipitate [37]. The average depth of the discolored area can be regarded as the chloride penetration depth. The chloride penetration depth was calculated by the white subareas. 

### 2.5. Free Chloride Content Measurement

After 28 FCTs, the specimens were baked under 60 °C for 2 days. The specimen divided by every 5 mm were drilled from the pure bending zone to measure the free chloride content. Samples were made into powder after drying. Powder below 0.15 mm was selected and baked at 60 °C for 1 day. Then, 2% concentration solution with 30 min mixing was prepared for the test.

## 3. Results

### 3.1. Flexural Capacity

Load-midspan deflection curve of the 28-day-cured specimen was plotted in Figure 3. The shadow area represents the experimental limits. The different deflection levels are marked with colored dotted lines, and the average of the stresses are indicated by black lines. As anticipated, ECC has demonstrated the deflection hardening behavior [35]. The average ultimate flexural strength is 13.83 MPa, corresponding to a deformation of 10.72 mm. The applied four deflection levels (DLs), namely 0, 0.1, 0.3 and 0.5, were determined as 0 mm, 1.07 mm, 3.22 mm and 5.36 mm, respectively. They correspond to the stress level of 0.59, 0.75 and 0.88, respectively.

### 3.2. Fluid Absorption

Figure 4 shows the change of the absorbed fluid over time. The absorption process can be divided into two regimes. Firstly, the specimens absorb fluid rapidly during the initial two days in which the absorbed water accounts for about 80% of the absorbed water. Afterwards, the absorption rate gradually slows down. Clearly, the water absorption increases with the applied deflection, as expected. As the deflection increases, microcracks develop within the specimen causing the specimen to absorb more solution [38]. Furthermore, the absorption increases with the salt solution concentration as salt added the saturation of the ECC [39]. The increment of the absorbed solution between 3% and 5% salt concentrations is smaller than that between 1% and 3% salt concentrations. This is in agreements with the finding reported in [40]. In terms of NaCl solution, 6% concentration leads to the maximum saturation.

Plain concrete absorb approximately 350 g/m^2^ after 7 days immersion in 3% NaCl solution, while the water absorption of ECC reaches 957 g/m^2^ under the same conditions. This is partly due to the interface formed between PVA fiber and matrix, and the −COOH and −OH functional groups in PVA dragging the water molecules forward [41]. Furthermore, as ECC consists of high volume fly ash, the pozzolanic reaction during immersion consumes a large amount of water [42]. At the same time, the addition of the PVA fiber also increases the open porosity, resulting in higher water absorption [43]. Therefore, the absorbed fluid weight of ECC is significantly larger than that of plain concrete.

### 3.3. RDEM

#### 3.3.1. Longitudinal RDEM

Figure 5 shows the evolution of longitudinal RDEM along the FTCs. The RDEM decreases with the accumulation of FTCs, indicating that damage or cracks are generated and accumulated inside the material due to the freezing and thawing, and similar results have been presented in [44]. In some cases, the transversal RDEM shows a slight increasing trend within the first 12 FCTs due to the recovered continuity of the materials. The hydration products produced by the ongoing hydration and pozzolanic reaction fills part of the FTCs-induced cracks or the interface between fiber and matrix [45]. However, there is no obvious spalling and damage on the surface, see Figure 6. Cracks were hardly observed in all the specimens, except for the combination with DL = 0.5. It indicates that the cracks produced by the ECC are very fine and may be healed during the freezing and sawing process under the low deflection. At high stress levels, white stuff is present on the crack surface of the specimen. It may be the formation of Ca(OH)_2_ and calcium carbonate [46]. Clearly, when the salt concentration is 1%, no change is found and even the specimen is under a relatively large DL. Specifically, after 28 FTCs, the RDEM only decreases 10% for DL = 0.5 in 1% salt solution. This indicates that ECC has a strong resistance to FTCs even under a relatively high stress level of sustained load. Similar results have been reported by [35]. When the salt concentration increases to 3%, a more severe loss of RDEM is observed. This is in accordance with the observation in plain concrete or mortar [47] as the glue-spall stress reaches the maximum at this pessimum salt concentration [24]. With no load, the loss of RDEM is about 5%. Nevertheless, the loss of RDEM is much lower compared with plain concrete [48] in which the loss of RDEM reaches 20% when exposed to a similar condition in just 16 cycles. This is because fibers can effectively resist the expansion pressure generated by the freezing. Meanwhile, the fiber introduces flaws into the matrix, which is beneficial for frost resistance [49].

The sustained load accelerates the deterioration process as it is shown that the loss of RDEM increases with DL. In case of 3% salt concentration, when DL = 0.5, the RDEM loss reaches 14.38%, which is 10.46% higher than the 3.92% of DL = 0. Moreover, the role of the DL on the deterioration is more significant for the solution with 3% salt concentration. This can be explained to the crack of brine ice formed in the pessimum salt concentration (3%) [24]. The RDEM losses are 2.39% and 9.73% in 1% and 5% salt solutions, which are 11.99% and 4.65% smaller than that of 5% salt solution (14.38%), respectively. Although specimens in 5% salt solution absorb more fluid, the pressure and the volume of solutions are decreased with increasing concentration [50]. Therefore, the most serious deterioration is observed in 3% salt solution.

#### 3.3.2. Transversal RDEM

Figure 7, Figure 8 and Figure 9 show the evolution of the transversal RDEM at various positions along the FTCs for 1%, 2% and 3% salt solutions, respectively. The flexural stress levels at positions A, B, C and D correspond to 0, 0.25, 0.5 and 1 of the applied maximum flexural stress at the pure bending zone. The loss of transversal RDEM is no more than 10%, which is obviously lower than the longitudinal one. In the same manner as the longitudinal one, the slight increment in the first 12 FTCs is observed for the transversal RDEM.

The loss of transversal RDEM after 28 FTCs and the local flexural stress is plotted in Figure 10. It clearly shows that the RDEM loss and flexural stress can be expressed as a linear relationship. The following equation was, therefore, proposed:(1)$$y=y_{0}+a\sigma$$
where *y* is the RDEM loss (%), *σ* is the flexural stress (MPa), *y*_0_ is the RDEM loss with no stress, a is the fitting constant denoting the influence of the flexural stress on the RDEM loss. The fitting shows satisfactory results as the determination coefficient (R^2^) is higher than 0.95 for all cases. The value of a is almost the same for the 1% and 5% salt solutions, indicating that the influence of flexural stress on the deterioration for the two concentration is more or less the same. Flexural stress is more significant for the transversal RDEM loss when the specimen is subjected to 3 % salt solution, i.e., the pessimum concentration. This is consistent with the observation of longitudinal RDEM. 

### 3.4. Chloride Penetration Depth

Figure 11a–c plots the measured chloride penetration depth at various positions after 28 FTCs. Regardless of the de-ice salt concertation, the further from the support, the higher penetration depth is observed. This can be attributed to the fact that the area further from the support is under greater sustained local stress level. Combined with the FTCs, more microcracks are created in the area under higher sustained stress. At the support, no significant change is found between specimens under different load levels. This indicates that the shear force at the plane has little contribution to the chloride ingress from the exposed surface. When the deflection level is 0.1 (stress level = 0.59), no significant increase is observed for all cases. This does not hold once the DL reaches 0.3. In terms of 1% NaCl solution, the penetration depth increases from 2.28 mm to 2.52 mm when the deflection level grows from 0.1 to 0.3 at the pure bending zone. The penetration depth increases to 2.83 mm when DL = 0.5.

When the chloride concentration is above 3%, no significant increment on the chloride penetration depth is observed. Specifically, at the position under zero flexural stress, the pentation depths are 2.08, 2.80 and 2.83 mm for 1%, 3% and 5% de-ice salt solution, respectively. Only 1% increment is observed when de-ice salt concentration increases from 3% to 5%, while the increment is 34% between 1% and 3% concentration. A more remarkable increase is observed when deflection is applied. The penetration depth for DL = 0.5 is 2.83 mm for 1% de-ice salt solution. It grows up to 5.51 mm and 6.06 mm for 3% and 5 % de-ice salt solution with 94.7% and 114.1% increment, respectively. When no load is applied, the chloride penetration depth after 28 FTCs using 5% NaCl solution is comparable with the one after 11 dry and wet cycles using the same salt solution [51]. 

Figure 12 presents the evolution of chloride penetration depth with the flexural stress level. An exponential equation was prepared to describe the relationship between them:(2)$$D=be^{c\sigma }$$

Here, *D* is the penetration depth of chloride (mm), *σ* is the flexural stress (MPa) and b and c are the fitting parameters. Parameter b represents the penetration depth under no load and c is used to represent the influence of flexural stress. The exponential equation shows satisfactory fitting results with determination coefficients higher than 0.9. Significant increment is observed for b and c when the salt concentration increases from 1% to 3%, while no big difference is found between 3% and 5% salt concentrations. It is worth mentioning that crack width becomes much larger with increasing the load level, the chloride may penetrate from the side of the crack. Therefore, the proposed relationship may not work. Nevertheless, the stress level has reached about 0.9 of the load capacity of the materials, which would merely happen for the structural design.

### 3.5. Free Chloride Content Profile

Figure 13 shows the free chloride content profile in the pure bending zone. The content of free chloride decreases with the depth perpendicular to the exposed surface and the slope becomes gentler. In terms of 1% salt concentration, the free chloride content is 0.22%, 0.26%, 0.28% and 0.32% under different deflection levels at 0–5 mm depth, respectively. The increments are 18.1%, 27.3% and 45.5% for DL = 0.1, 0.3 and 0.5, respectively, compared with DL = 0.

The free chloride content at a depth of 0–5 mm does not show significant difference between the 3% and 5% de-ice salt solutions. The free chloride content is 1.29 % and 1.48% for 3% and 5% de-ice salt solutions under no load, respectively. It increases to 1.96% and 1.95% for the 3% and 5% de-ice salt solutions when DL = 0.5, respectively. Regarding the depth of 15–20 mm, when no load is applied the free chloride content for 3% salt solution is smaller than that of 5% salt solution. Once the DL is applied, the free chloride content for 3% salt solution is higher than 5% salt solution. The difference increases with the DL. Specifically, it is 6.25%, 10.53% and 22.2%, for DL = 0.1, 0.3 and 0.5, respectively. 

The critical chloride content (Ccrit) is generally considered as 0.4% by the weight of cement (% cem wt) in Europe and North America [52], although other values, e.g., 0.2% cem wt and 0.6% cem wt are also considered [53]. Herein, 0.4% cem wt is used as the key values of the steel [54,55]. Clearly, no depassivation of the steel is expected, even under a large deflection level, when subjected to FCTs under 1% de-ice salt solution. In terms of 5% salt solution, the thickness of cover should be no less than 20 mm when DL = 0.5 is applied. This does not apply for the 3% salt solution as more serious deterioration occurs in such a concentration. Nevertheless, when DL < 0.3, no depassivation of the steel at a depth of 20 mm is expected within 28 FCTs.

## 4. Conclusions

In this paper, the effect of coupled salt frost attack and sustained load on ECC was investigated. The loss of RDEM, along with the FTCs and chloride penetration depth and free chloride content after 28 FTCs, was measured. The conclusions are as follows.

Both the sustained load and salt concentration (no more than 5%) result in higher fluid absorption. The water absorption distributes in the range of 600–1800 g/m^2^. The increment rate decreases with the salt concentration.A 3% NaCl solution leads to the largest deterioration under salt frost with or without load. A linear relationship can be used to describe the relationship between locally sustained flexural stress and RDEM loss after FTCs. The load level is most significant on the RDEM loss in 3% NaCl solution (i.e., the pessimum concentration). The RDEM loss rate in the 3% NaCl solution is almost 50% greater than others.The chloride penetration depth increases with the locally sustained flexural stress. A significant increment of chloride penetration depth is observed when salt concentration increases from 1% to 3%. Almost 100% increment occurs in the pure bending zone, while no big difference is found between 3% and 5% salt concentrations.After FTCs, the free chloride content at a depth of 0–5 mm does not show remarkable difference for the 3% and 5% salt solutions. Regarding depth of 15–20 mm, when no load is applied, the free chloride content for 3% salt solution is smaller than that of 5% salt solution. Once the sustained load is applied, the free chloride content for 3% salt solution is higher than the 5% salt solution. The difference increases with the DL.When the ECC is subjected to the FTCs under 1% de-ice salt solution, no depassivation of the steel is expected even under a large deflection level. In terms of 3% and 5% salt solution, the thickness of cover should be no less than 20 mm when DL = 0.5 is applied.

The behavior of ECCs subjected to the coupled sustained flexural load and salt frost was investigated, albeit limited to the meso-scale. Further studies, including the change of the microstructure and chemical phases at the micro-scale, are required. 

## Figures and Tables

**Figure 1 materials-16-00165-f001:**
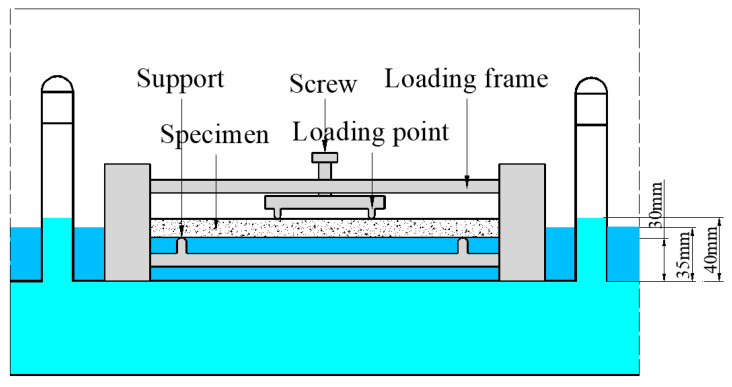
Schematic view of the test configuration.

**Figure 2 materials-16-00165-f002:**
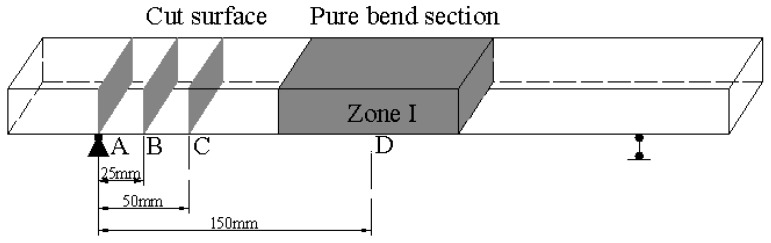
Specimen cutting position.

**Figure 3 materials-16-00165-f003:**
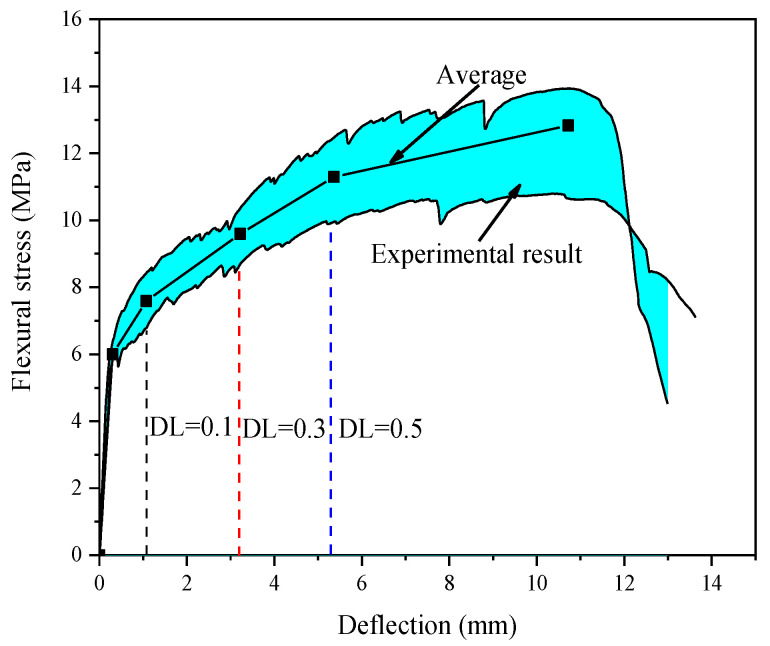
Applied deflection level.

**Figure 4 materials-16-00165-f004:**
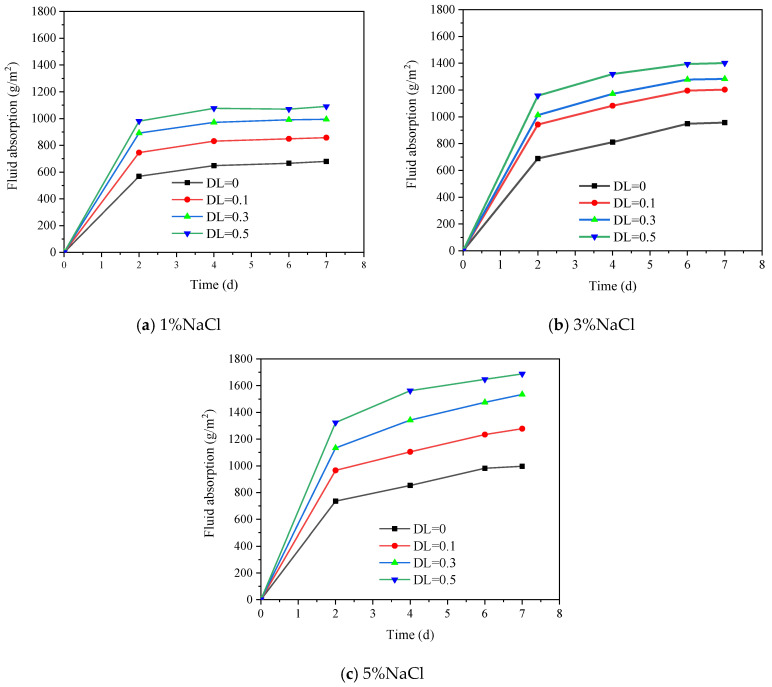
The relationship between water absorption and time for different salt solution concentrations.

**Figure 5 materials-16-00165-f005:**
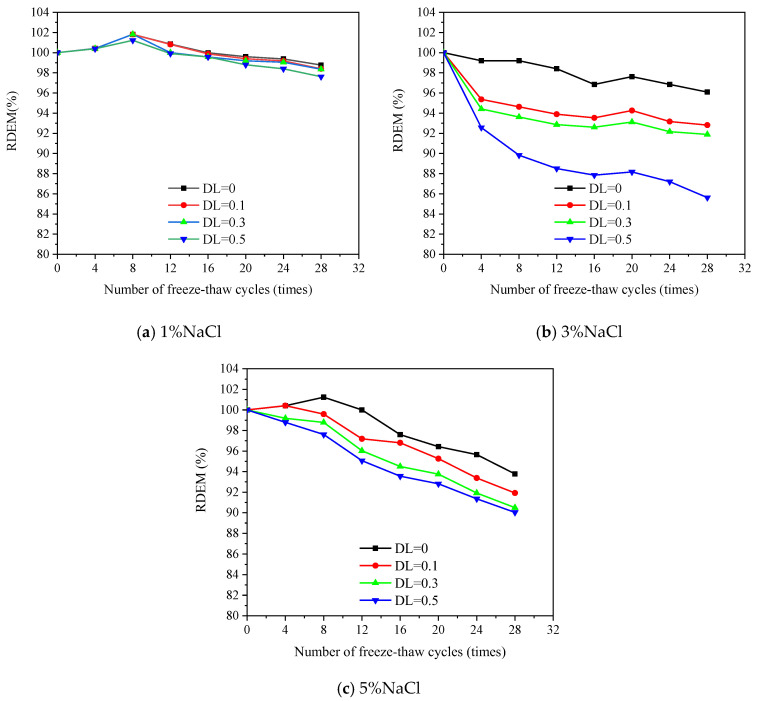
Evolution of longitudinal RDEM along the FTCs under various salt concentrations.

**Figure 6 materials-16-00165-f006:**
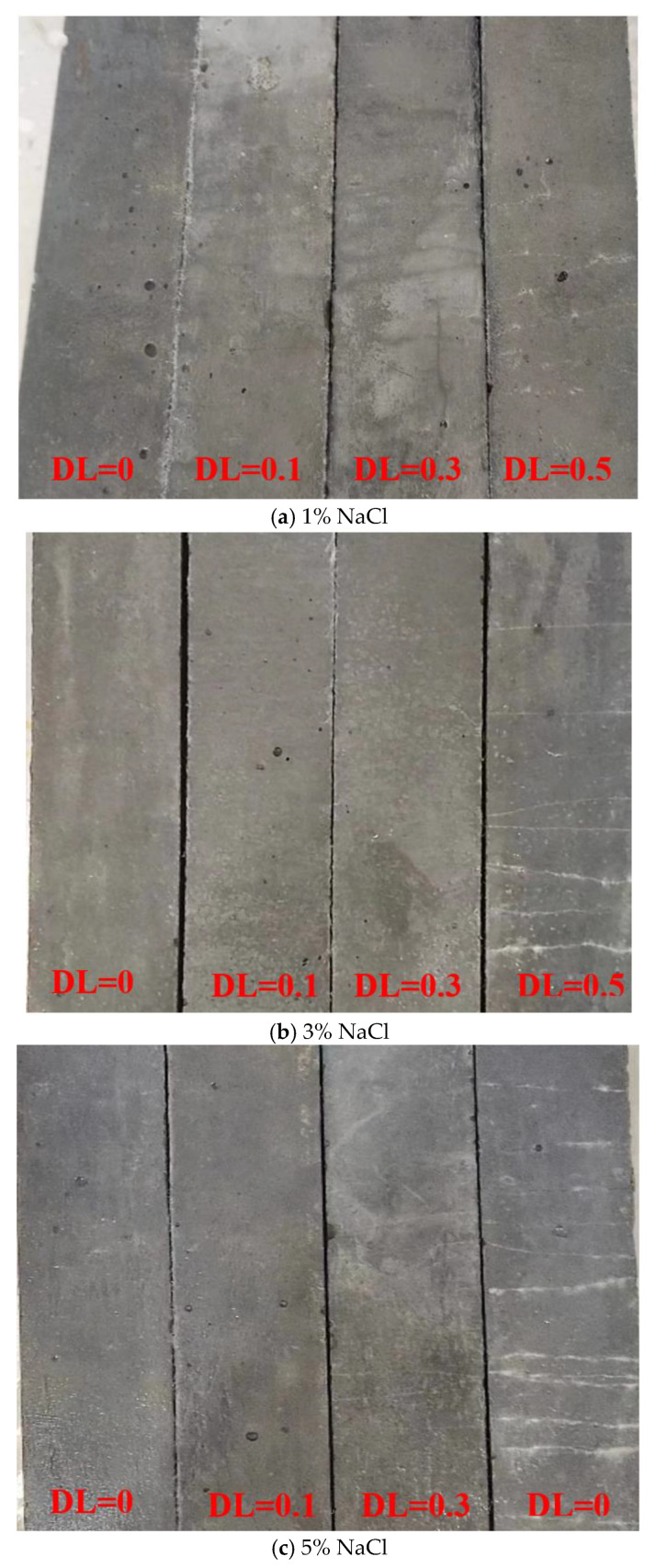
Surface of specimens after 28 FTCs under different salt concentrations.

**Figure 7 materials-16-00165-f007:**
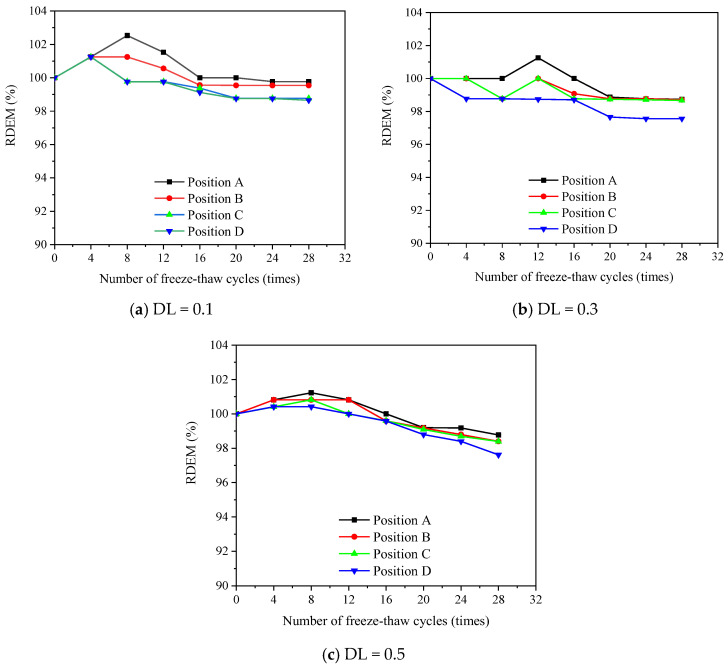
Measured transversal RDEM for specimens under different DL and FCTs subjected to 1% salt solution.

**Figure 8 materials-16-00165-f008:**
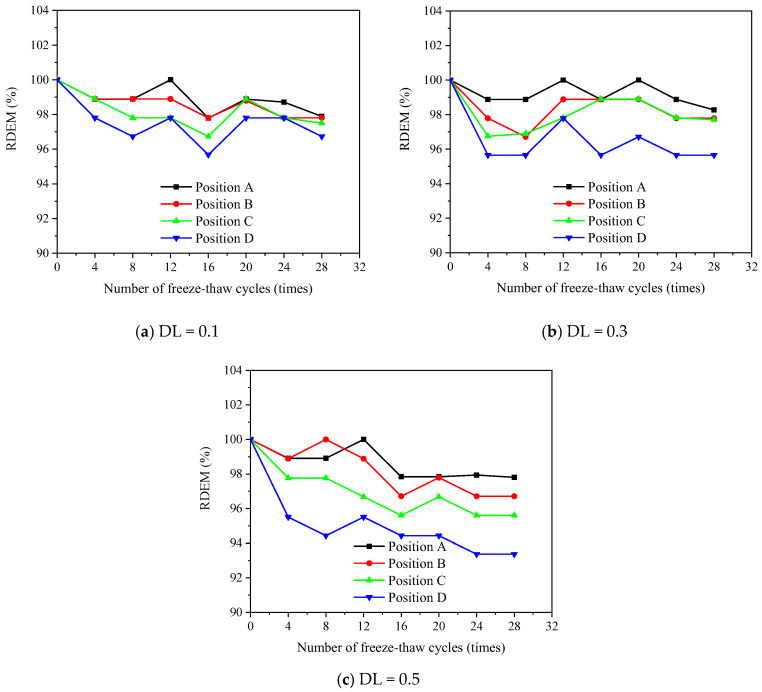
Measured transversal RDEM for specimens under different DL and FCTs subjected to 3% salt solution.

**Figure 9 materials-16-00165-f009:**
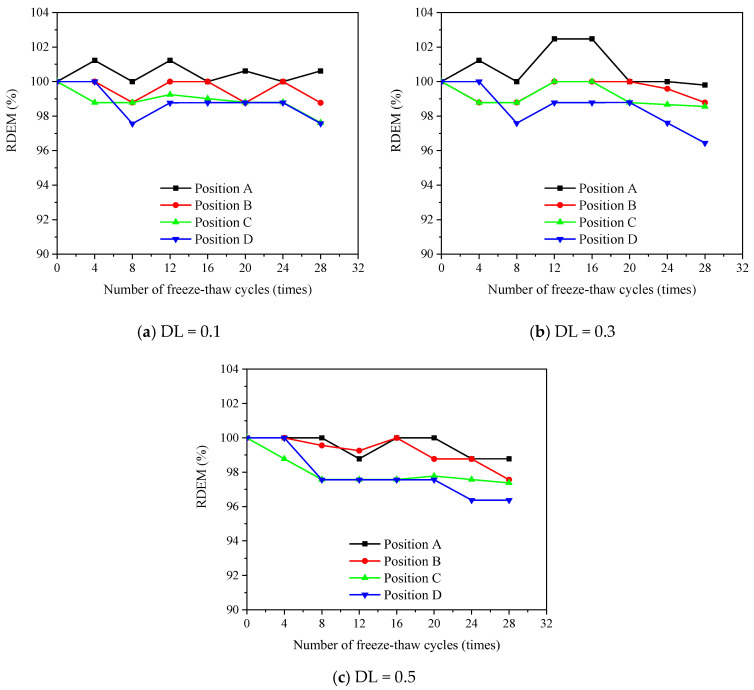
Measured transversal RDEM for specimens under different DL and FCTs subjected to 5% salt solution.

**Figure 10 materials-16-00165-f010:**
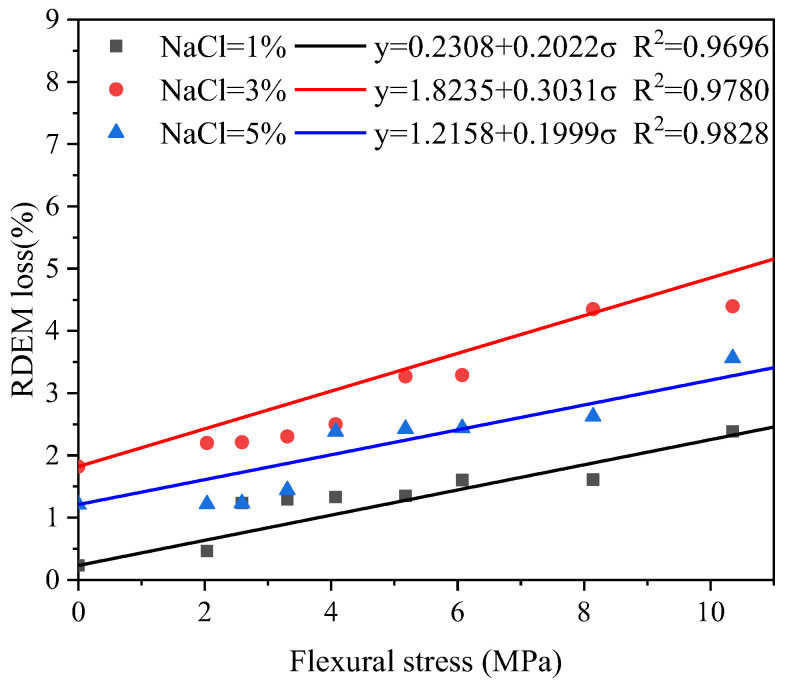
Influence of local flexural stress on the transversal RDEM loss.

**Figure 11 materials-16-00165-f011:**
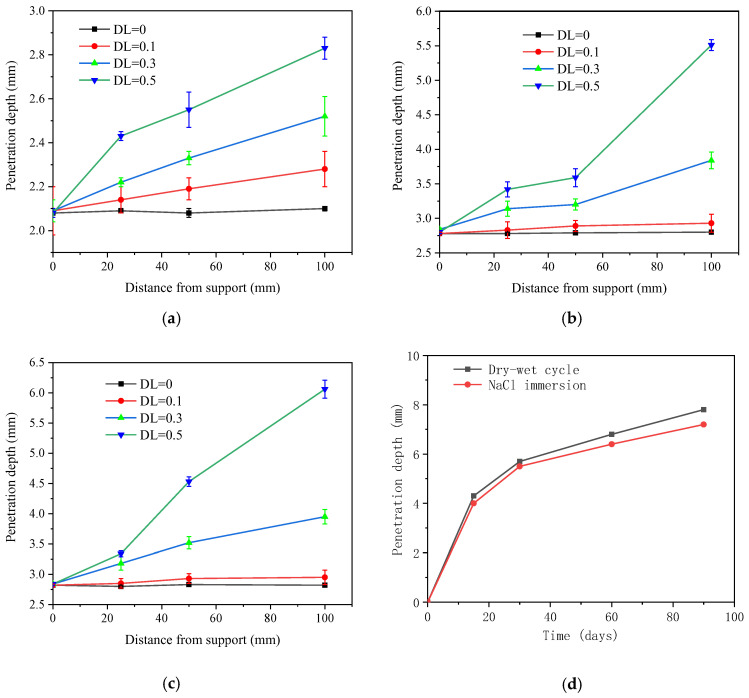
Chloride penetration depth after 28 FCTs under various conditions. (**a**) 1% NaCl, (**b**) 3% NaCl, (**c**) 5% NaCl, (**d**) Chloride penetration depth in 5% NaCl solution [51].

**Figure 12 materials-16-00165-f012:**
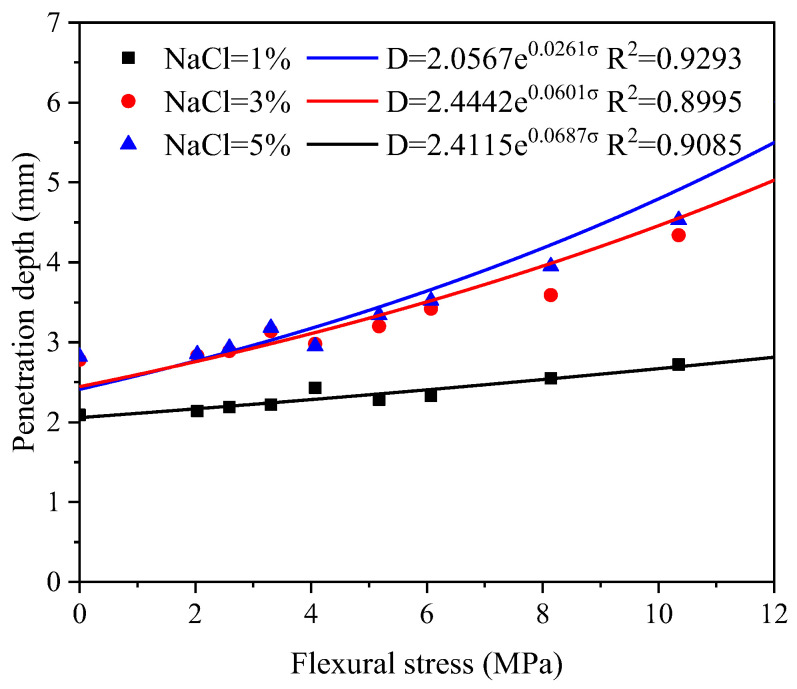
Evolution of chloride penetration depth along flexural stress level after 28 FCTs.

**Figure 13 materials-16-00165-f013:**
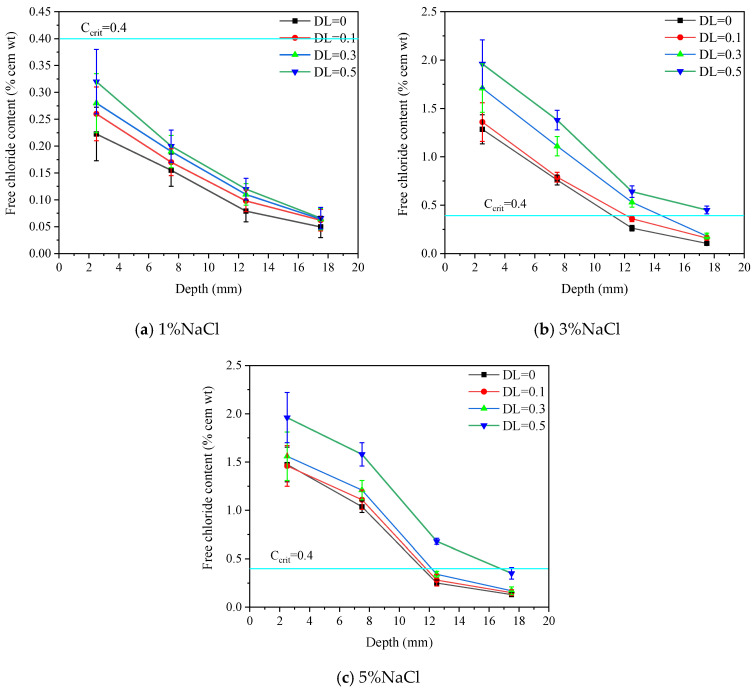
Free chloride ion content after 28 FTCs under various conditions.

**Table 1 materials-16-00165-t001:** Chemical compositions of cement and fly ash (wt.%).

Oxides	CaO	SiO_2_	Al_2_O_3_	Fe_2_O_3_	MgO	Na_2_O	K_2_O	MnO	TiO_2_	SO_3_	P_2_O_5_
Cement	63.21	18.48	6.74	3.45	3.24	0.17	0.53	0.27	0.35	3.16	0.16
Fly ash	3.34	49.66	35.97	5.77	0.63	0.62	0.93	0.99	0.04	1.12	0.28

**Table 2 materials-16-00165-t002:** The mixture proportions of PVA-ECCs (kg/m^3^).

Cement	Fly Ash	Quartz Sand	Water	Superplasticizer	VMA	PVA Fiber	W/b
593	712	474	339	4.74	0.55	26	0.26

## Data Availability

The data presented in this study are available upon request from the corresponding author.

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
