# Peer review of "Behavior of Engineered Cementitious Composites (ECCs) Subjected to Coupled Sustained Flexural Load and Salt Frost"

_materials, 2022, doi:10.3390/ma16010165_

Round 1
Reviewer 1 Report
In the paper, authors investigate the behavior of engineered cementitious composites (ECCs) under coupled sustained flexural loading and salt frost attack.
The current work has lack of enough literature support. In my opinion this paper needs major revision.
Comment (1): The novelty of the investigation needs more explanation because FRP sheets with spike anchors has been widely studied in literature.
Comment (2): Presented results should be compared to the existing ones.
Comment (3): The results need more deep explanation.
Reviewer 2 Report
REVIEW
on article
Behavior of engineered cementitious composites (ECCs)
subjected to coupled sustained flexural load and salt frost
Yonghao Li, Ning Zhang, Renjuan Sun, Yanhua Guan, Lemin Liu, Hongzhi Zhang, Changjin Tian,
Branko Šavija, Yifeng Ling
SUMMARY
The article submitted for review is relevant. The behavior of engineered cement composites subjected to coupled sustained flexural load and salt frost is presented.
The purpose of this research is to study the behavior of cements under an interesting complex effect - mechanical and temperature. The authors conducted in-depth laboratory studies, analyzed frost resistance, water absorption, mechanical characteristics, and established relationships between the data obtained and input factors. Thus, their study is very important from the point of view of engineering science and materials science. This article has some value but should be finalized with the following comments.
COMMENTS
1. The authors should indicate what kind of paper they submitted (I mean Article) and remove other types (line 1).
2. The presented Abstract does not reflect the essence of the study. The authors should revise it. The authors immediately proceed to the purpose of the study without identifying the problems, that is, it is not clear for what purposes and the solution of what problems this was carried out. The background should be reflected at the beginning of the Abstract.
3. In addition, the methodology is listed in great detail with specific levels of parameter variation in terms of salt concentration and deflections of bending elements but does not reflect the quantitative characteristics of the obtained scientific result itself. Thus, the abstract should be reworked in terms of issues and scientific results.
4. The Introduction section given by the authors is too concise. Authors must submit a detailed literature review, from which a scientific problem should follow, in accordance with it, the goal and tasks are set. In the current version of the manuscript, the authors have not done this and must complete the introduction.
5. Literary review is superficial. It is not clear how sources 9-11 or 13-16 differ. It is necessary to approach this issue in more detail and present it to the reader. In addition, it is necessary to touch upon the behavior of not only concretes, like cement composites, but also other stone materials, which, of course, are subjected to freezing in salts and some kind of mechanical action at the same time. There are many studies on this topic, the authors should work on a literature review.
6. In section 2.1 "Materials", justify the selected materials on line 86. The chemical composition of the cement was presented; however, the mineralogical composition was not presented. It should also be added after Table 1.
7. Figure 2 shows interesting diagrams that need further explanation.
8. Figure 3 is not entirely clear; it should be explained.
9. The graphs in Figure 4 and Figure 5 are not informative, and they are not well explained.
10. The photographs in Figure 6 are presented in poor quality, they should be presented in a higher quality. Similar remarks to Figures 7-13.
11. Discussion of the results obtained is not presented in the form required in journals of this level. A detailed comparative analysis of the obtained results with the results obtained earlier by other authors should be carried out. Perhaps a general summary graph is needed that will reflect points with data obtained by the authors for other materials, or for other concentrations of solutions, or at other temperatures. The authors must give a detailed comparison, and only then will the reader understand the contribution of the authors to science and scientific novelty, and the new knowledge that has been obtained or the existing ideas that have been developed.
12. The Conclusions are presented very superficially and require a generalizing addition in the form of prospects for the development of this study and the practical applicability of the results obtained, as well as information about new ideas received.
13. The list of references looks impressive - 62 sources. But not all of these sources are fully explored in the text of the article. The authors should systematize and align the list of references and the text of this article.
14. In general, the article is interesting and scientifically developed, but some things are not enough in it, for example, it would be useful to present an SEM analysis of the microstructure of the materials under study. The analytical component should also be strengthened. The article is quite promising but needs a significant revision. After completion, the article must be sent for re-reviewing.
Reviewer 4 Report
Please further elaborate on the novelty of your work in abstract.
The presented introduction is pretty modest. Please include a brief but critical review regarding the conducted research studies in the introduction. Accordingly, please include a brief summary of using cementitious materials on mechanical properties of concrete using the article titled Predicting the Compressive Strength of Concrete Containing Binary Supplementary Cementitious Material Using Machine Learning Approach.
It is recommended to add a section “research significance” and highlight the main contribution of your findings.
Moreover, please further elaborate on improving the resistance of structures against corrosion using GFRP material following the outcomes of the study titled Effect of Fiber Reinforced Polymer Tubes Filled with Recycled Materials and Concrete on Structural Capacity of Pile Foundations.
Please include the statistical characteristics of the reported results based on RDEM and FTC in Figure 5.
Please explain the potential of crack development patterns during the conducted experiments accordingly to Figure 6.
Please include a comparative analysis on the main parameters that can considerably affect the results of this study and include a compelling discussion in this regard.
Please state the main shortcomings during the conducted experiments.
Please mention the verification procedure of the conducted analysis and the performed experimental tests.
Please revise the conclusion with a quantified-based approach.
Round 2
Reviewer 1 Report
The authors replied to all comments
Reviewer 2 Report
All my comments were taken into account and appropriate corrections were done in the article's text.
I recommend the article for publishing.
Reviewer 3 Report
The authors properly addressed my comments and significantly improved the manuscript quality. I have no further comment.
Reviewer 4 Report
N/A